Lateralized alpha oscillations are irrelevant for the behavioral retro-cueing benefit in visual working memory

Mössing Wanja A. 1 2 moessing@wwu.de
Busch Niko A. 1 2
1 Institute of Psychology, University of Münster , Münster , Germany
2 Otto Creutzfeldt Center for Cognitive and Behavioral Neuroscience, University of Münster , Münster , Germany
Barnhart Anthony
Electronic publication date: 2020 Jun 23
Publication date: 2020
Volume: 8
Electronic Location ID: e9398
Received 2020 Feb 19; Accepted 2020 May 30
Copyright: © 2020 Mössing and Busch
Copyright year: 2020
Copyright holder: Mössing and Busch
License: This is an open access article distributed under the terms of the Creative Commons Attribution License, which permits unrestricted use, distribution, reproduction and adaptation in any medium and for any purpose provided that it is properly attributed. For attribution, the original author(s), title, publication source (PeerJ) and either DOI or URL of the article must be cited.
License URL: https://creativecommons.org/licenses/by/4.0/

Keywords: Neural oscillations, Alpha rhythm, Alpha lateralization, Visual attention, Vision, Working memory, Hemispheric lateralization, Retro-cue, Short term memory

Funding: The authors received no funding for this work.

==============================
The limited capacity of visual working memory (vWM) necessitates the efficient allocation of available resources by prioritizing relevant over irrelevant items. Retro-cues, which inform about the future relevance of items after encoding has already finished, can improve the quality of memory representations of the relevant items. A candidate mechanism of this retro-cueing benefit is lateralization of neural oscillations in the alpha-band, but its precise role is still debated. The relative decrease of alpha power contralateral to the relevant items has been interpreted as supporting inhibition of irrelevant distractors or as supporting maintenance of relevant items. Here, we aimed at resolving this debate by testing how the magnitude of alpha-band lateralization affects behavioral performance: does stronger lateralization improve the precision of the relevant memory or does it reduce the biasing influence of the irrelevant distractor? We found that it does neither: while the data showed a clear retro-cue benefit and a biasing influence of non-target items as well as clear cue-induced alpha-band lateralization, the magnitude of this lateralization was not correlated with any performance parameter. This finding may indicate that alpha-band lateralization, which is typically observed in response to mnemonic cues, indicates an automatic shift of attention that only coincides with, but is not directly involved in mnemonic prioritization.

Introduction

Working memory is the ability to maintain and process information that is no longer physically present. It acts as the mind’s main work space and is a core element of human cognition underlying many critical functions (cf., Baddeley, 2015) such as general intelligence (Conway, Kane & Engle, 2003), attention (Oberauer, 2019), or reading comprehension (Tighe & Schatschneider, 2016).

Working memory is strictly capacity-limited (Woodman & Fukuda, 2018). In particular, working memory for visual information (visual working memory, vWM) is limited to approximately 3–4 items (Luck & Vogel, 1997; Mance & Vogel, 2013). According to so-called slot models, this limit arises due to a fixed number of discrete slots or stores, each of which can maintain only one item. Any items exceeding the number of available slots cannot be represented in vWM. By contrast, resource models (Ma, Husain & Bays, 2014; Schneegans & Bays, 2016) hold that vWM has a limited processing resource, which can be flexibly distributed across an arbitrary number of items. The more items are maintained, the less resources are available per item, reducing the quality of the items’ representations. While these models disagree on the architecture of vWM and on the exact cause of capacity limitations, they both highlight that without prioritization of relevant over irrelevant information, the available capacity may be insufficient for maintaining all information with the desired quantity or quality.

A common way to manipulate prioritization experimentally is to show an array of multiple items, and inform participants about which target items are to be encoded using a visual cue. This prioritization has been shown to enhance vWM performance, as indicated by better memories for cued targets as compared to non-cued distractor items (Ma, Husain & Bays, 2014). Cues presented before or simultaneously with the items (so-called pre- and simu-cues, respectively) help prioritizing relevant information before it is fully encoded. Surprisingly, even cues presented after the stimuli have disappeared (so-called retro-cues) improve performance (Griffin & Nobre, 2003; Rerko, Souza & Oberauer, 2014; for a review, see Souza & Oberauer, 2016). Given that this retroactive improvement occurs well after all stimuli have been fully encoded, it indicates a shift of vWM resources to either improve the relevant (Rerko, Souza & Oberauer, 2014) or inhibit/remove the irrelevant representation(s) (Souza et al., 2014; Lewis-Peacock, Kessler & Oberauer, 2018; Myers, Stokes & Nobre, 2017). However, the neuronal mechanisms underlying this shift are not fully understood.

A candidate neuronal mechanism for this shift of vWM resources are occipito-parietal oscillations in the alpha frequency range (∼7–14 Hz). By convention, an increase in alpha power is referred to as event-related synchronization, implying that greater power observed at the scalp is due to more synchronized activity in neuronal populations. Numerous studies have shown that greater synchronization in the alpha band is related to physiological, perceptual, and cognitive inhibition or inactivity (Pfurtscheller, 1992; Klimesch, Sauseng & Hanslmayr, 2007; Sauseng et al., 2009; Jensen & Mazaheri, 2010; Bonnefond & Jensen, 2012; Händel, Haarmeier & Jensen, 2011; Weisz et al., 2014; Iemi et al., 2017; Dougherty et al., 2017). Paradoxically, greater WM load—presumably associated with greater cognitive activity—is also associated with greater alpha synchronization, which has been explained with a greater need for prioritization and for inhibiting potentially distracting information (Klimesch et al., 1999; Jensen, 2002; Busch & Herrmann, 2003; Jokisch & Jensen, 2007). Consistent with this interpretation, when targets and distractors are presented in different hemifields, alpha synchronizes during the maintenance interval specifically over the hemisphere processing distractors and desynchronizes over the hemisphere processing the targets. This lateralization of alpha power has been observed in studies using both pre-cues and retro-cues (Poch, Campo & Barnes, 2014; Myers et al., 2015; Clayton, Yeung & Cohen Kadosh, 2018; Bacigalupo & Luck, 2019; Schroeder, Ball & Busch, 2018). However, a lateralized topography could, in principle, reflect both: synchronization/inhibition of populations coding irrelevant information and desynchronization/excitation of populations coding relevant information (Noonan et al., 2016; Bacigalupo & Luck, 2019; Foster & Awh, 2019). Moreover, several studies have reconstructed features of the memorized target from the spatial pattern of alpha power during the maintenance interval (Sprague & Serences, 2015; Foster et al., 2016; Ester, Nouri & Rodriguez, 2018; Foster & Awh, 2019). This finding might suggest that alpha oscillations are involved in the maintenance of the targets rather than in the inhibition of distractors. In sum, it is currently unclear whether cue-induced lateralization of alpha power during a vWM maintenance interval is associated with target representation, distractor inhibition, or both.

The goal of this study was to test whether alpha lateralization during the maintenance interval is a neuronal mechanism supporting the retro-cueing effect in vWM, and whether it accomplishes this via facilitation or inhibition. On each trial, we presented two lateralized items and presented a retro-cue during the maintenance interval, thereby defining one item as the target and the other as the distractor (Fig. 1). Participants then reported the orientation of the target item in a continuous delayed estimation task. Data from this task were analyzed with a modeling approach, yielding parameter estimates for the precision of the relevant target, for guessing, and for bias induced by the irrelevant distractor. We predicted that trials with a retro-cue yield more precise memories of the cued target and less interference with the irrelevant item than trials without a cue. Moreover, we predicted that the topography of alpha power lateralizes by decreasing power in the hemisphere contralateral to the cued target. Finally, we tested the relationship between the magnitude of retro-cue induced alpha lateralization and behavioral performance: does stronger lateralization facilitate the precision of the relevant memory, or does it inhibit the biasing influence of the irrelevant memory?

Figure 1 The task in detail (scale adjusted for illustration).

Trials started with central fixation on a black–white-striped fixation symbol for 500 ms (verified by eyetracking). Then, two lateralized items (1 black; 1 white) were displayed for 1,000 ms, followed by a first delay interval of 500 ms during which participants memorized both items. Thereafter, a retro-cue was presented for 1,000 ms on 2/3 of all trials by changing the color of the fixation symbol entirely to the color of the to-be-remembered target (100% valid). On the remaining non-cued trials, the regular fixation symbol remained on screen for 1,000 ms. The cued target (or both items, in case of non-cued trials) had to be maintained for another 500 ms during a second delay interval. Finally, participants reproduced the orientation of the target indicated by the response wheel’s color.

Materials and Methods

We report how we determined our sample size, all data exclusions (if any), all manipulations, and all measures in the study (“21-word solution”, see Simmons, Nelson & Simonsohn, 2012).

Participants

Forty participants with normal or corrected-to-normal vision were recruited online and via campus-wide advertisements, and were compensated for participation with course credit or money (8 €/h). One participant did not return for a second session and was therefore excluded from all analyses, leaving 39 participants (25 female; mean age = 24.82 years, SD = 4.76; 26 right eye dominant; 35 right-handed). The study was approved by the ethics committee of the faculty of psychology and sports science, University of Münster (#2016-23-NB). All participants gave their written consent to participate. The sample was determined a priori based on sample sizes in similar studies.

Apparatus

Recordings took place in a medium-lit, sound-proof cabin. Participants placed their heads on a chin-rest and could adjust the height of the table to be seated comfortably. Stimuli were generated using Matlab (2016b) (The Mathworks Inc., 2016) and the Psychophysics Toolbox 3 (Brainard, 1997; Pelli, 1997; Kleiner, Brainard & Pelli, 2007). The experiment was controlled via a computer running Windows 10, equipped with an Intel Core i5–3330 CPU, a 2 GB Nvidia GeForce GTX 760 GPU, and 8 GB RAM. The experiment was displayed on a 24″ Viewpixx/EEG LCD Monitor with 120 Hz refresh rate, 1 ms pixel response time, 95% luminance uniformity, and 1920 × 1080 pixels resolution (www.vpixx.com). Distance between participant eyes and the monitor was approximately 86 cm. Participants responded using a wired Logitech RX250 optical USB mouse (www.logi.com).

Stimuli and procedure

All stimuli were presented on a gray background (50 cd/m2). A central black-and-white fixation symbol (0.6° diameter; 0.1 and 100 cd/m2, respectively; see Thaler et al., 2013) was displayed throughout the trial, except when it was temporarily replaced by the retro-cue and during the report task (see below). Target stimuli were two pseudo-randomly oriented arrows (length: 3.85°, linewidth: 0.2°, headwidth: 0.49°) with a minimum orientation difference of 10° and a maximum difference of 120°, excluding orientations along the cardinal axes. Target orientations were pseudo-randomly sampled so that they were approximately uniformly distributed from 1–360° across trials. One arrow was presented in each hemifield, centered on the vertical meridian at 5° eccentricity. On each trial, one arrow was black, the other white. The combination of target color and hemifield was counterbalanced between participants. That is, for one half of the participants the left arrow was always black, while for the other half the left arrow was always white (and vice versa for the right arrow).

Although a higher set size might have increased the retro-cueing benefit, a set size of only two items (one target, one distractor) was necessitated by the research question: does alpha lateralization support the retro-cueing benefit by facilitation of the target or by inhibition of the distractor? First, precision of target reports must not be limited by interference from unreported targets. Second, errors must be unequivocally quantified in terms of attraction or repulsion relative to the distractor, which is impossible with more than a single distractor.

Participants initiated a trial by fixating on the fixation symbol for 500 ms. Next, the two target arrows appeared for 1,000 ms. Participants were instructed to always encode both arrows. The targets were followed by a first delay interval of 500 ms, during which both target arrows had to be retained. On retro-cue trials, the fixation symbol then turned into a cue by changing color either to black or white for 1,000 ms, informing participants to memorize only the arrow of the corresponding color with 100% validity, rendering the other arrow a distractor. By contrast, on non-cued trials the fixation symbol remained on screen unchanged for 1,000 ms, requiring participants to keep memorizing both arrows. The cue’s directionality thus depended on the color association with the target arrows. The retro-cue was followed by a second delay interval of 500 ms. A comparison of the EEG before and after cue onset thus reflects the change in vWM caused by the cue. Finally, participants were tested on an initially blank response wheel on which they had to reproduce the target orientation by moving a sample arrow with the mouse. The response wheel’s color matched the color of the to-be-reproduced arrow. To avoid biasing participants with a pre-defined orientation, the arrow only appeared after an initial mouse-movement of at least 25 pixels into a desired direction was made. The response wheel was always presented centrally rather than at the location of the to-be-reported target, so that EEG lateralization towards the memorized target during the delay interval was not confounded with attention towards the expected location of an upcoming stimulus. Responses were self-paced and participants were encouraged to respond as accurately as possible. Responses were followed by a feedback display, showing the correct response next to the given response for a duration between 1,250 and 1,750 ms, uniformly jittered across trials. Figure 1 displays the full sequence of one trial.

Participants completed 576 trials with valid retro-cues (cued side was pseudo-randomized across trials, 50% for each hemifield). In addition, they completed 288 non-cued trials without a cue. Cued and non-cued trials were presented in separate blocks of 48 trials each. Blocks were interleaved with self-paced breaks. A total of 864 trials was split evenly between two sessions of approximately 90 minutes each.

Eye-tracking

Eye-movements were monitored using a desktop-mounted Eyelink 1000+ infrared based eye-tracking system (www.sr–research.com) set to 1,000 Hz sampling rate (monocular). Pupil detection was set to centroid fitting of the dominant eye. The eye-tracker was (re-)calibrated using a nine-point calibration grid at default locations.

Participants were required to keep fixating on the fixation symbol throughout the trial, except when they were giving the report and during the feedback interval. To ensure steady fixation and to avoid ocular artifacts or preferential encoding of one of the targets, a trial was aborted and repeated at the end of the respective block whenever participants blinked or gaze was outside of a 3° radius around the fixation symbol. The eye tracker was recalibrated at the start of each block and whenever participants lost fixation for more than 3,000 ms.

Analysis of behavioral performance

To analyze the effect of retro-cueing on behavioral performance, responses were quantified for each trial as a response error: the angular difference between the target orientation and the reported orientation. In order to test if reports were systematically biased by the irrelevant distractor stimulus, the signs of the response errors were recoded such that positive values indicated errors in the direction of the distractor, and negative values indicated errors in the opposite direction.

The distribution of errors was then fitted with the MemToolbox for Matlab (Suchow et al., 2013) using a standard mixture model (Zhang & Luck, 2008) with an additional bias parameter. According to this model, each trial is drawn from a circular normal distribution when the participant has a memory representation, and drawn from a uniform distribution otherwise (representing random guessing). The model has three free parameters: κ, representing the precision of the participant’s memory representations, is the concentration parameter of the circular normal distribution (inversely related to dispersion)—the higher this concentration, the more precise the representation in memory. μ is the mean of the Von-Mises distribution; a deviation of μ from zero represents a systematic memory bias towards or away from the orientation of the distractor. g is the probability of the item not being in memory when tested (i.e., the proportion of random guesses), which is represented by the height of the uniform distribution.

The model’s probability density function is thus given by: f(x;g,μ,κ)=(g)1360+(1−g)Φ(x;μ,κ)

where Φ is the von Mises density: Φ(x;μ,κ)=ekcos(x−μ)360I0(k)′

where I0 is the modified Bessel function of order zero (Fougnie, Suchow & Alvarez, 2012). Models were fitted for each individual using maximum likelihood estimation (MLE) for each of the three parameters, separately for the two conditions (no-cue; cue).

EEG acquisition and preprocessing

EEG was recorded with a Biosemi Active Two EEG system with 65 Ag/AgCl electrodes (www.biosemi.nl), set to 1,024 Hz sampling rate. A total of 64 electrodes were arranged in a custom made montage with equidistant placement (‘‘Easycap M34’’; www.easycap.de), which extended to more inferior areas over the occipital lobe than the conventional 10–20 system (e.g., Oostenveld & Praamstra (2001), see comparison in Supplemental Material). An additional external electrode was placed below the left eye (IO1).

EEG data were preprocessed using Matlab, R2018a (The Mathworks Inc., 2018) and the open-source EEGlab toolbox (Delorme & Makeig, 2004) with the EYE-EEG (Dimigen et al., 2011), Cleanline (Mullen, 2012), and SASICA (Chaumon, Bishop & Busch, 2015) extensions. Data were resampled to 512 Hz, high-pass filtered at 0.5 Hz, low-pass filtered at 100 Hz, and notch-filtered at 50 Hz. Subsequently, data were re-referenced to a robust reference (Bigdely-Shamlo et al., 2015). Vertical and horizontal electrooculograms were derived from two electrodes above and below the left eye, and two electrodes at the lateral canthi of both eyes, respectively. The EEG data were then co-registered with the eye-tracking data, using TTL triggers sent simultaneously to both devices (Dimigen et al., 2011).

The combined data were segmented into epochs from 1,000 ms before target onset until 4,500 ms after target onset. Epochs that were aborted due to eye- or mouse-movements were rejected and the remaining epochs co-registered with behavioral data. Epochs with irregular artifacts where automatically detected using a combination of threshold criterion, maximum slope criterion, joint probability, and kurtosis (see code for specific values). Regular artifacts were corrected using Independent Component Analysis (ICA). Artefactual components were automatically pre-selected using the SASICA extension (Chaumon, Bishop & Busch, 2015) and rejected after an additional manual inspection. Finally, the data of both sessions were combined into a single dataset per participant.

Data were transformed into the time-frequency domain using complex Morlet wavelets for 30 logarithmically spaced frequency bands from 2 to 40 Hz with the number of cycles increasing from 1 to 10 as a function of frequency. To analyze the effect of retro-cues, the resulting power values were baseline corrected relative to the first delay interval, that is, 500 ms before cue onset. For each participant, power was transformed to a dB scale by computing baseline corrected power pwc as: pwct,f,i=10∗log10(pwt,f,ibslf,i)

where pwt,f,i is raw power at each time point t, frequency f and trial i, and bslf,i is power averaged across the baseline interval for each f and i. Thereby, pwc highlights the relative change elicited by the retro-cue and attenuates effects of earlier stimulation. All subsequent analyses were performed on pwc.

EEG analysis

Sample level alpha lateralization: cluster based permutation test

Memorization of a lateralized target is expected to induce a lateralized response in the alpha band following the cue. To test for alpha-band lateralization in the signal averaged across trials, a non-parametric cluster based permutation test (CBPT; Maris & Oostenveld, 2007) was conducted to test ipsilateral vs. contralateral alpha-band power (i.e., mean pwc over all trials and 7–14 Hz) for the time interval between retro-cue onset and probe onset. The CBPT was implemented in FieldTrip (Oostenveld et al., 2011) with 1,000 randomizations and a cluster-level alpha threshold of 0.05. Specifically, a two-tailed dependent-samples t-test was calculated for each sample (electrode × time). Thereafter, temporally adjacent samples which were significant at α = 0.05, and involved at least two spatially adjacent electrodes, were selected as clusters. For each cluster, a test statistic was computed by taking the sum of t-values within that cluster (tsum). Thereafter, condition labels (i.e., ipsi- vs. contralateral) were randomly permuted across samples 1,000 times and clusters were computed in the same manner for each of the 1,000 random partitions. A null permutation distribution was then generated by taking the maximum cluster statistic from each of the 1,000 random partitions. A cluster in the real data was considered significant if its cluster statistic lay below the 2.5th, or above the 97.5th percentile of this null distribution. This procedure successfully controls the false alarm rate in multiple comparison problems.

Quantification of lateralization in single trials

Next, we tested how the single-trial strength of lateralization was related to each trial’s behavioral performance. As single trial data are inherently noisy and individual participants’ lateralization patterns can differ substantially from the grand average pattern, each participant’s lateralization was quantified based on their individual lateralization template using a classification approach.

Specifically, trials were classified based on how well their topographical patterns of alpha power could predict the cued hemifield. First, each trial’s pwc was averaged over the alpha-band (7–14 Hz) and the time-window found significant according to the CBPT, separately for all 64 electrodes (i.e., one topography per trial). A difference topography was then calculated by subtracting the average of all right-cued trials from the average of all left-cued trials, representing that participant’s lateralization template. Subsequently, the correlation of each trial’s topography with that template was computed (see Fig. 2 for a schematic overview). Consequently, single trials showing a topography typical for left-cued trials will show a negative correlation with the template, while typical right-cued trials will correlate positively (see Schurger, Marti & Dehaene, 2013, for a similar approach). To evaluate the consistency of topographies across trials, we tested how accurately trials could be classified as left-cued or right-cued based on these single-trial topographical correlations. To this end, a receiver operating characteristic (ROC) curve analysis was computed using the perfcurve.m function from Matlab’s Statistic Toolbox. The ROC relates correct and incorrect classification across different classification thresholds. Importantly, the ROC analysis yielded an optimal cut-off value for each participant, that is, a topographical correlation value that provided the most accurate classification. Accordingly, any trials with a correlation higher than this threshold were classified as “right-cued” and trials with a correlation lower were classified as “left-cued”. Trials in the no-cue condition were similarly classified, using the probed hemifield instead of the cued hemifield to determine a trial’s true category. Finally, by comparing the classified cue condition with the trials’ true condition, trials were then categorized as showing “consistent lateralization” or “inconsistent lateralization”.

Figure 2 Classification procedure.

(A) Participants’ individual templates were calculated as follows. First, the time-frequency area of interest was averaged over time and frequency per electrode, yielding one value per electrode & trial (i.e., one topography per trial). Thereafter, topographies of right-cued and left-cued trials were separately averaged, producing one topography per condition. Finally, the topography for right-cued trials was subtracted from the one for left-cued trials, resulting in a template topography. (B) Single-trial classification: correlations between all trials’ topographies and the template topography were calculated, resulting in a single correlation value per trial. Next, a ROC curve was computed, which yielded an individual optimal cut-off value, that is, a topographical correlation value that provided the most accurate classification. Finally, trials with a correlation above this threshold were classified as left-cued, whereas trials with a lower correlation were classified as right-cued.

To test if classification accuracy was above chance, the classification was repeated 1,000 times with permuted condition labels. A permutation null distribution was then generated from the classification accuracies of the 1,000 random partitions. The true classification accuracy was then compared to the permutation null distribution of random accuracy scores to compute a p-value: p=nk/ntot+1

where nk is the number of random accuracies that are higher than the true test accuracy score, and ntot the number of random partitions.

In sum, this analysis quantifies how well the topography of a single trial matches the lateralization pattern of all trials. Importantly, this procedure does not require a fixed selection of electrodes for each participant, based on the grand average topography or on a priori regions of interest.

Relationship between lateralization and performance

We then analyzed how single-trial lateralization correlated with single-trial behavioral performance. If lateralization was functionally relevant for vWM maintenance, trials categorized as having “consistent lateralization” are expected to yield better performance than those with “inconsistent lateralization”. As a split into just two groups may dilute statistical power, the sets of “consistent” and “inconsistent” trials were additionally parted using a set-wise median split on correlation with the template, yielding four groups of trials: “consistent trials with strong correlation”, “consistent trials with moderate correlation”, “inconsistent trials with moderate correlation”, and “inconsistent trials with strong correlation”. Subsequently, standard mixture models were fitted separately for these four categories, and their parameter estimates were compared between trial classes1 .

In addition, we tested whether participants with more “consistent overall lateralization” performed better than participants with “inconsistent overall lateralization” (Händel, Haarmeier & Jensen, 2011). Specifically, standard mixture models were fitted separately for each participant and their parameter estimates were compared using classification accuracy as a continuous predictor, representing “(in)consistency” of an individual’s lateralization.

Importantly, comparing mixture model parameter estimates between “consistent” and “inconsistent” lateralization patterns allowed us to test if lateralization affects the precision of target representations (κ parameter) or the bias imposed by the distractors (μ parameter).

Furthermore, it is important to demonstrate that both lateralization and behavioral measures show sufficient across-participants variability, and that this variability is indicative of reliable inter-individual differences rather than measurement error. To this end, we computed, separately for each mixture model parameter, the correlation between the first and second recording session across participants. This analysis showed that parameter estimates and alpha-band lateralization were variable across participants, and highly reliable within participants, thereby validating the analysis of correlations between lateralization and behavior (see Supplemental Information S4 for details).

Statistical inference

Differences in parameter estimates (κ, μ, and g) were statistically analyzed using the R statistical programing language (R Core Team, 2019), with the afex library for frequentist inference (Singmann et al., 2019), and the BayesFactor library for analogous Bayesian tests (Morey & Rouder, 2018). Reported p-values are Bonferroni–Holm corrected for multiple comparisons, where appropriate (Holm, 1979).

To test for an effect of retro-cues on behavioral performance, each parameter estimate was compared between cued and non-cued trials using separate two-tailed dependent samples t-tests and two-sample Bayes factors (ttestBF, default 2/2 priors; see Morey & Rouder, 2011).

To test for an effect of irrelevant distractors on the memories of relevant targets, a two-tailed one-sample t-test, and a one-sample Bayes factor were computed to assess whether bias (μ) parameter estimates were different from zero.

To test for an effect of single trial lateralization on performance, each parameter estimate was analyzed with a separate 2 × 4 linear mixed effects model (LMM) with fixed factors cue (cued, non-cued), and lateralization (“consistent trials with strong correlation”, “consistent trials with moderate correlation”, “inconsistent trials with moderate correlation”, “inconsistent trials with strong correlation”), and a random intercept for ID. Similarly, the effect of participants’ overall lateralization was tested using LMMs with the fixed factors cue (cued, non-cued), and lateralization (continuous), and a random intercept for ID. P-values were obtained using Kenward–Rogers approximation.

For the same designs, Bayes factors for all fixed effects and interactions were computed against the null hypothesis that no factor, except the random factor id, has an effect—that is, against the non-informative model parameter ∼ id (generaltestBF, default 2/2 priors; see Morey & Rouder, 2011). The resulting Bayes factors can be directly interpreted as evidence favoring one model over another. Note that the subscripts of reported Bayes factors are used to indicate the favored model: BF10 favors the alternative hypothesis, whereas BF01 favors the null hypothesis (i.e., BF01 is simply 1/BF10). A Bayes factor BF10 larger than one for the model parameter ∼ lateralization thus reflects evidence in the data for a relation of lateralization to that particular standard mixture model parameter, whereas a Bayes factor BF01 larger than one reflects evidence against that effect.

Results

Behavioral retro-cue effect

Not surprisingly, raw response errors did not differ significantly between retro-cued and non-cued trials at such a small set size (Md = 0.21, 95% CI [−0.18, 0.60], t(38) = 1.08, p = 0.286; see Fig. 3A). However, the maximum likelihood estimates (MLEs) of the mixture model parameters (Fig. 3B) showed that precision κ was significantly higher for retro-cued trials (Mean difference (Md) = 5.40, 95% CI [3.13, 7.66], t(38) = 4.83, p < 0.001, d = 0.35). This result is much more likely under the assumption that retro-cues increase precision (BF10 = 942.09), providing extreme evidence against the null hypothesis that retro-cues did not influence precision. Guessing g approached zero in both conditions, reflecting that guessing is rarely necessary at a maximum set size of two items. Nevertheless, g was significantly higher for no-cue trials (Md = − 0.01, 95% CI [−0.01, 0.00], t(38) = −3.07, p = 0.012, d = −0.40). This result is about nine times more likely under the assumption that retro-cues reduce guessing (BF10 = 9.25). Bias μ was significantly above zero (M = 0.41, 95% CI [0.10, 0.72], t(38) = 2.65, p = 0.012, d = 0.42; BF10 = 3.58), indicating that reported target orientations were systematically biased in the direction of the irrelevant target. However, this bias did not differ between the two cue conditions. In fact our data provide moderate evidence in favor of the null hypothesis that retro-cues did not influence bias (BF01 = 3.91).

Figure 3 Distribution of response errors for retro-cued and non-cued trials.

Individual mean errors in degree are displayed along with sample kernel density estimates. Boxes show Cosineau-Morey (Morey, 2008) within-participants standard errors and the corresponding whiskers are 95% CIs. To test the strength of a potential response bias, response errors were recoded, such that positive values indicated errors in the direction of the distractor, and negative values indicated errors in the opposite direction. (B–D) Distribution of maximum likelihood estimates (MLEs) of the parameters of a Standard Mixture Model with bias. Error bars indicate 95% Cosineau-Morey CIs.

Sample level alpha lateralization

After selecting the alpha frequency band (7–14 Hz) as a priori frequency range of interest, the CBPT revealed a difference between power ipsilateral compared to power contralateral to the cued target (tsum = 4,301.66, CI [4,301.659, 4,301.663], p < 0.001). Significant lateralization effects were most pronounced in the time range from 300 to 1,070 ms over posterior electrodes (Fig. 4). In non-cued trials, no significant difference was found between electrodes located ipsilateral compared to contralateral relative to the target that was probed in the upcoming test.

Figure 4 (A) Temporal evolution of alpha-power (7–14 Hz) in non-cued and retro-cued trials.

For retro-cued trials, power is shown separately for ipsi- and contralateral electrodes. Shaded areas are standard errors. Power is averaged over the electrodes found significant in the cluster based permutation test (CBPT). The label “cluster/topo” highlights the time interval during which ipsi- and contralateral power differed significantly (see topography in (C)). (B) Time-frequency resolved difference between ipsi- and contralateral activity in retro-cued trials (contra – ipsi) for the electrodes highlighted in (C). The white rectangle highlights the time-frequency region of the significant cluster. (C) Topography of lateralization in the alpha-band, averaged over the time interval 500–1,000 ms post retro-cue onset. Highlighted electrodes are part of the significant cluster for at least 50% of this time interval. All displayed data are averaged across participants.

Single trial alpha-based classification

Single trials in the retro-cue condition were classified as cued-left and cued-right based on their correlation with each participant’s right vs. left difference topography. Classification accuracy ranged from 0.56 to 0.83 and was, on average, significantly above chance (M = 0.612, SE = 0.007, p < 0.001).

Relating alpha lateralization and performance

Within participants: single trial fluctuations of alpha power

By contrasting single trials that were correctly and incorrectly classified, we used classifier predictions as a proxy for “consistent single-trial lateralization” and “inconsistent single-trial lateralization”, respectively, to test for effects of single-trial alpha-band lateralization on single-trial behavioral performance. To disentangle the effect of single-trial lateralization on vWM, we fitted four separate mixture models for “consistent” and “inconsistent” trials, where each set was additionally split by the median absolute correlation with the classification template. Figure 5A summarizes the estimated parameters.

Figure 5 (A–C) Distribution of maximum likelihood estimates (MLEs) of the parameters of a Standard Mixture Model with bias, separately fit for “consistent trials with strong correlation”, “consistent trials with moderate correlation”, “inconsistent trials with moderate correlation”, and “inconsistent trials with strong correlation”.

Error bars indicate 95% Cosineau-Morey CIs. (D–F) Distribution of maximum likelihood estimates (MLEs) of the parameters of a Standard Mixture Model with bias. For visualization and comparability to the single trial data, participants are split by quartiles of classification accuracy. Error bars indicate 95% Cosineau-Morey CIs.

For each mixture model parameter, we computed a 2 × 4 linear mixed effects model (LMM) with fixed factors cue (cued, non-cued), and lateralization (“consistent trials with strong correlation”, “consistent trials with moderate correlation”, “inconsistent trials with moderate correlation”, “inconsistent trials with strong correlation”). While this analysis replicated the effect of cueing on precision and guessing, single-trial lateralization had no effect on any model parameter (see Table 1).

Table 1 Results of linear mixed models with fixed factors cue (cued, non-cued), and lateralization (“good trials with strong correlation”, “good trials with mild correlation”, “weak trials with mild correlation”, “weak trials with strong correlation”), and random interecept for ID.

P-values are Bonferroni–Holm corrected for multiple comparisons.

Parameter	Effect	F	df1	df2	pholm	
κ	Cue	36.06	1	266	<0.0001***	
	Lateralization	0.02	3	266	>0.99	
	Lateralization × cue	0.20	3	266	>0.99	
g	Cue	6.85	1	266	0.03*	
	Lateralization	0.79	3	266	>0.99	
	Lateralization × cue	0.62	3	>0.99	>0.99	
μ	Cue	0.48	1	266	0.03*	
	Lateralization	1.61	3	266	0.57	
	Lateralization × cue	0.17	3	>0.99	>0.99	
Notes:

* p < 0.05.

*** p < 0.001

In addition, we computed Bayes factors for the same design. In line with the LMMs, the precision and guessing MLEs are most likely under a model with just a main effect of cue (i.e., κ ∼ cue + id; BF10 = 2,050,467 and g ∼ cue + id; BF10 = 3.39). The results are less likely under any model including the factor lateralization. In fact, all models including just lateralization as fixed effect (i.e., estimate ∼ lat. + id) perform worse than the null model (see Table 2), providing evidence in favor of the null hypothesis that single-trial lateralization did not influence vWM.

Table 2 Results of Bayes factor analysis with fixed factors cue (cued, non-cued), and lateralization (“good trials with strong correlation”, “good trials with mild correlation”, “weak trials with mild correlation”, “weak trials with strong correlation”) against the null model parameter ∼ID, with ID as random factor.

BF10 represents evidence in favor of H1 over H0, whereas BF01 represents evidence in favor of H0 over H1 (i.e., BF01 = 1/BF10).

Parameter	Model	Bayes factor	
κ	κ ∼ cue + id	BF10 = 2,050,467	
	κ ∼ lat. + id	BF01 = 66.64	
	κ ∼ cue + lat. + id	BF10 = 30,400.29	
	κ ∼ cue + lat. + cue:lat. + id	BF10 = 1,272.554	
g	g ∼ cue + id	BF10 = 3.39	
	g ∼ lat. + id	BF01 = 25.37	
	g ∼ cue + lat. + id	BF01 = 7.66	
	g ∼ cue + lat. + cue:lat. + id	BF01 = 113.08	
μ	μ ∼ cue + id	BF01 = 6.36	
	μ ∼ lat. + id	BF01 = 8.40	
	μ ∼ cue + lat. + id	BF01 = 53.77	
	μ ∼ cue + lat. + cue:lat. + id	BF01 = 1,357.97	

Between participants: individual differences in alpha power

By contrasting participants using classification accuracy, we used classification as a proxy for “lateralization consistency”, ranging from “consistent overall lateralization” to “inconsistent overall lateralization”, to test for effects of a participant’s overall alpha-band lateralization on their behavioral performance. To disentangle the effect of individual overall lateralization on vWM, we fitted separate mixture models for each individual participant.

Figure 5B illustrates the distribution of parameter estimates across participants, split by quartiles of classification accuracy. Interestingly, the figure suggests that the retro-cue benefit for precision κ is marginally higher for participants with higher classification accuracies. This trend, however, was not statistically significant.

A LMM with factors cue (cued, non-cued), and lateralization (continuous) replicated the effect of cue on precision and guessing, but showed no effects of lateralization on any mixture model parameter (see Table 3).

Table 3 Results of linear mixed models with fixed factors cue (cued, non-cued), and lateralization (continuous, participant-wise accuracy scores), and random intercept for ID.

P-values are Bonferroni–Holm corrected for multiple comparisons.

Parameter	Effect	F	df1	df2	pholm	
κ	Cue	24.81	1	37	<0.0001***	
	Lateralization	1.66	1	37	0.63	
	Lateralization × cue	3.50	1	37	0.21	
g	Cue	9.20	1	37	0.01*	
	Lateralization	1.18	1	37	0.84	
	Lateralization × cue	0.03	1	37	>0.99	
μ	Cue	0.84	1	37	>0.99	
	Lateralization	0.68	1	37	>0.99	
	Lateralization × cue	0.86	1	37	>0.99	
Notes:

* p < 0.05.

*** p < 0.001.

In addition, we computed analogous Bayes factors. In line with the LMM results, the precision and guessing MLEs are most likely under a model with just a main effect of cue (i.e., g ∼ cue + id; BF10 = 9 and κ ∼ cue + lat. + id; BF10 = 704.76), providing strong evidence in favor of the alternative hypothesis that the retro-cue reduced guessing and improved precision. The results are less likely under any model including the factor lateralization. In fact, all models including just lateralization as fixed effect (i.e., estimate ∼ lat. + id) perform worse than the null model. For bias μ, the null model outperformed all alternative models, similar to the within-participant analysis (see Table 4).

Table 4 Results of Bayes factor analysis with fixed factors cue (cued, non-cued), and lateralization (continuous, participant-wise accuracy scores) against the null model parameter ∼ID, with ID as random factor.

BF10 represents evidence in favor of H1 over H0, whereas BF01 represents evidence in favor of H0 over H1 (i.e., BF01 = 1/BF10).

Parameter	Model	Bayes factor	
κ	κ ∼ cue + id	BF10 = 704.76	
	κ ∼ lat. + id	BF01 = 1.39	
	κ ∼ cue + lat. + id	BF10 = 605.84	
	κ ∼ cue + lat. + cue:lat. + id	BF10 = 615.50	
g	g ∼ cue + id	BF10 = 9	
	g ∼ lat. + id	BF01 = 1.67	
	g ∼ cue + lat. + id	BF10 = 5.78	
	g ∼ cue + lat. + cue:lat. + id	BF10 = 1.77	
μ	μ ∼ cue + id	BF01 = 2.94	
	μ ∼ lat. + id	BF01 = 1.94	
	μ ∼ cue + lat. + id	BF01 = 6	
	μ ∼ cue + lat. + cue:lat. + id	BF01 = 12.21	

Discussion

Previous research has demonstrated that the capacity of visual working memory (vWM) is strictly limited (Woodman & Fukuda, 2018; Luck & Vogel, 1997; Mance & Vogel, 2013). This limit makes prioritization of relevant over irrelevant information highly important for sensible human action and cognition. Cues about the future relevance of items are therefore highly beneficial for the quality of memory representations during the maintenance interval and, in turn, for accuracy at test (Ma, Husain & Bays, 2014). Cues can be presented while target stimuli are still on-screen, thereby prioritizing the relevant items for encoding through selective attention to their physical locations. Interestingly, cues can even be presented when the items are no longer physically present, and thus after encoding has finished. The benefit afforded by such retro-cues demonstrates that prioritization can be achieved by selective attention to internal memory representations in addition to external screen locations (Souza & Oberauer, 2016).

A candidate neuronal mechanism for this shift of vWM resources are alpha oscillations: across many different paradigms, strong alpha power has been associated with a suppression of neural, perceptual and cognitive processing (Jensen & Mazaheri, 2010; Clayton, Yeung & Cohen Kadosh, 2018). Moreover, when relevant and irrelevant items are represented in different hemispheres, the power of alpha oscillations lateralizes such that power increases over the hemisphere representing the irrelevant information relative to the hemisphere representing the relevant information (Sauseng et al., 2009). However, it is largely unknown how exactly this lateralization is involved in prioritization and whether it is a neuronal correlate of the inhibition of irrelevant items (Myers, Stokes & Nobre, 2017) or of the preferential encoding of relevant items (Foster & Awh, 2019). Moreover, only few studies have specifically investigated the role of alpha lateralization induced by retro-cues for the retro-cueing benefit. Therefore, this study aimed at testing predictions derived from these opposing views regarding the association between the magnitude of retro-cue induced alpha lateralization and behavioral performance: while the former predicts that stronger lateralization improves the precision of the relevant memory, the latter predicts that stronger lateralization reduces the biasing influence of the irrelevant memory.

We presented two lateralized items on each trial and presented a retro-cue during the maintenance interval, thereby defining one item as the target and the other as the distractor (Fig. 1). Participants then reported the orientation of the target item. Critically, the retro-cue was purely symbolic (its color indicated the color of the to-be-memorized item) and the probe stimulus used for the continuous report task was located in the center of the screen. Thus, any lateralization induced by the retro-cue can be attributed to a shift of vWM resources towards the internal representation of the cued item, rather than to an external location on screen or to the anticipated location of a lateralized probe.

Retro-cue benefit and distractor bias

Our behavioral analysis (Fig. 3) showed a clear retro-cue benefit: compared to a control condition without cues, retro-cues increased the precision of the memorized item and reduced the proportion of guessing, confirming previous studies on the retro-cue benefit (Souza & Oberauer, 2016). While a retro-cue benefit for guessing has been more frequently reported in the literature than a benefit for precision, effects on precision are more prevalent in studies with small set sizes such as ours. This is possibly due to unreliable estimation of the precision parameter when performance is low due to high memory load (cf. Lawrence, 2010; Souza & Oberauer, 2016).

Furthermore, the representation of the relevant item was systematically biased toward the irrelevant item, although this attraction bias was not affected by the retro-cue. Several studies have reported a similar attraction bias, whereby the orientation of a memorized target line was biased towards an irrelevant distractor presented either before the target (Wildegger et al., 2015) or during the retention interval (Lorenc et al., 2018; Rademaker et al., 2015). By contrast, several other studies have found a repulsive bias for line orientation (Blakemore, Carpenter & Georgeson, 1970) and aspect ratio of ellipses (Sweeny, Grabowecky & Suzuki, 2011) under conditions with minimal memory requirements. Moreover, Czoschke et al. (2019) found repulsion for the memorized motion direction of two simultaneously presented random dot kinematograms. This condition resembles the uncued condition in our study, in which we found the strongest attraction bias, pointing to an inconsistency between studies. On the other hand, Czoschke et al. (2019) found attraction when the two stimuli were presented sequentially and the first stimulus was to be reported. Assuming that participants prioritized the first stimulus within a sequence, this condition is more consistent with the task and results in the cued condition in our study. Complicating this picture further, Bae & Luck (2020) found that the direction in which the reported orientation of one of two successive stimuli is biased depends on the stimuli’s similarity: while repulsion was found for similar orientations, attraction was found for dissimilar orientations. Both biases were reduced when a cue instructed participants to prioritize one of the stimuli. Is it possible that the apparent inconsistency between studies is due to a neglect of the items’ (dis)similarity? While we are not in a position to answer this question for previous studies, we re-analyzed the behavioral data of the present study by analyzing the memory error for the reported item as a function of the angular similarity between target and distractor (see Fig. S3 in supplemental analysis). In contrast to Bae & Luck (2020), our analysis showed attraction for most magnitudes of target-distractor (dis)similarity, and showed no evidence for repulsion when target and distractor had similar orientations, resembling results reported in Wildegger et al. (2015). In sum, it appears that the direction of the bias exerted by one stimulus on another depends on a host of factors such as the relevant stimulus feature (random dot motion, line orientation, etc.), stimulus timing (concurrent or sequential), memory requirements, and whether or not one of the items is prioritized. Importantly, whatever the determinants of the bias’ direction may be, our results clearly show that participants were able to use the retro-cue to prioritize memorization of the relevant item, and that the irrelevant item nonetheless exerted a distractive influence on that memory representation.

No association between alpha lateralization and retro-cue benefit

The cluster based permutation test revealed a significant lateralization of alpha-band power following the retro-cue (Fig. 4): power decreased over the contralateral (i.e., target-processing) hemisphere relative to the ipsilateral (i.e., distractor-processing) hemisphere. This result replicates previous reports of alpha-band lateralization induced by retro-cues (Poch, Campo & Barnes, 2014; Van Ede, Niklaus & Nobre, 2017; Van Ede, 2018; Schneider et al., 2019).

The coincidence of retro-cue induced alpha-band lateralization and the behavioral retro-cueing benefit has conventionally been interpreted as a causal involvement (Poch, Campo & Barnes, 2014; Schneider et al., 2019), indicating a “mnemonic role” for alpha oscillations (Van Ede, 2018; Ester, Nouri & Rodriguez, 2018). This role could be relevant either for directly coding the memorized information, or for other processes indirectly facilitating its coding, for example, by changes in the excitatory state of the neuronal populations tuned to the target location. Such a change from “latent” to “active” storage might improve feature binding and reduce noise, and thereby improve memory quality of the target memory (Myers, Stokes & Nobre, 2017). Critically, such interpretations are mostly based on the comparison of lateralization and behavior across experimental conditions. However, coincidence is not necessarily indicative of a causal relationship, and alpha-band lateralization could be just a by-product of the neural processes underlying the behavioral retro-cueing benefit. While strong evidence for a causal effect of neural processes on behavior is generally hard to establish without invasive methods, a prerequisite for a causal link is at least a covariance between lateralization and behavior. Specifically, fluctuations in the extent of lateralization across trials or across participants should go along with fluctuations in behavioral performance. However, only few studies have analyzed the covariation of retro-cue induced lateralization and performance.

To test for a relationship between lateralization and performance across trials, we computed a single-trial measure of lateralization by computing how strongly each trial’s topographical pattern resembled the participant’s average lateralization pattern (Schurger, Marti & Dehaene, 2013). This lateralization measure was robust enough to decode the cued hemifield on a trial-by-trial basis (Fig. 2). However, behavioral performance (i.e., precision, guessing, and bias) was not significantly better on trials with “consistent” lateralization compared to trials with “inconsistent” lateralization (Tables 1 and 2) as shown by both conventional null hypothesis testing and a Bayes factor analysis. Additionally, we analyzed the relationship between lateralization and performance across participants. This analysis tested for the possibility that some people might be more capable than others to adapt the excitatory state of neuronal populations underlying the relevant memory representation, making them more likely to enjoy the retro-cueing benefit. To this end, we quantified each participants’ lateralization strength as the accuracy at which the cued hemifield could be decoded from their topographical pattern. However, behavioral performance was not significantly better for participants with “consistent” lateralization compared to participants with “inconsistent” lateralization (Tables 3 and 4).

While we quantified alpha-band lateralization using a classification-based approach, most previous studies have quantified lateralization by contrasting ipsilateral vs. contralateral electrodes. In order to compare our results to more “conventional” analyses, we also analyzed the correlation between participants’ behavioral retro-cueing benefits and their average alpha lateralization quantified as the ipsilateral-contralateral difference. The results were qualitatively very similar to the results of our classification-based analysis (see Supplemental Analyses S2 for details), showing no correlation between lateralization and retro-cueing benefit. Moreover, to assure that the classification-based quantification of alpha lateralization indeed captures more information than more conventional analyses, we compared classification accuracy using all electrodes with accuracy using just a limited set of electrodes which is typically used for obtaining the ipsilateral-contralateral difference. In brief, the comparison confirmed that cue condition could be classified more accurately based on the set of all electrodes (see Supplemental Analyses S1 for details), thereby confirming the validity of our classification approach. In sum, our failure to find an association between alpha-band lateralization and the retro-cue benefit does not seem to hinge upon any specific analysis procedure.

Our findings are largely consistent with previous studies that analyzed the correlation between alpha-band lateralization and accuracy. Neither Myers et al. (2015) nor Günseli et al. (2019) found a significant correlation across participants between alpha-band lateralization and accuracy. Using a paradigm even more similar to ours, Van Ede, Niklaus & Nobre (2017) showed only a weak (r ∼ 0.05) but significant correlation between single-trial contralateral alpha oscillations and response times, but not accuracy. In sum, although retro-cues clearly induced alpha-band lateralization and simultaneously improved vWM performance, our results provide evidence against a direct mnemonic role of alpha lateralization in a retro-cueing paradigm.

This negative finding is surprising for several reasons. First, an association between cue-induced lateralization and memory accuracy has at least been implied in several studies on the retro-cue benefit (Poch, Campo & Barnes, 2014; Van Ede, 2018; Schneider et al., 2019). Second, correlations between alpha-band power and accuracy have been observed in other sensory modalities (Spitzer & Blankenburg, 2012; Backer, Binns & Alain, 2015), and using pre-cues instead of retro-cues (Sauseng et al., 2009). Moreover, estimates of neuronal memory representations can be reconstructed from retro-cue induced alpha-band power, and these memory reconstructions reflect the behavioral retro-cueing benefit (Ester, Nouri & Rodriguez, 2018). However, while our results provide strong statistical evidence against an association between alpha lateralization and the retro-cue benefit, we acknowledge that these null findings should be interpreted with caution. While we found no evidence for an association using a variety of analysis strategies, it is difficult to make any claims about the absence of an association beyond the present study. Specifically, using a paradigm that yields a stronger behavioral retro-cue benefit or stronger alpha-band lateralization, and using more sensitive analysis techniques might reveal such an association after all.

Functional relevance of alpha-band lateralization

We would like to point out that this study was carefully designed to make sure that lateralization was indeed a result of memory prioritization in response to the retro-cue (albeit not the mechanism implementing prioritization), and to rule out alternative interpretations (see Van Ede, 2018). First, the stimulus display was always bilateral, and lateralization emerged only after retro-cue presentation. Thus, lateralization was not confounded by lateralized stimulation at encoding. Second, the retro-cue was purely symbolic: its color indicated the color of the to-be-memorized item. We opted against arrows as cues, which are asymmetrically shaped and highly overlearned stimuli. Thus, lateralization was not due to exogenous shifts of attention towards a lateralized point on the cue’s shape, nor due to automatic attentional orienting to the screen location pointed to by the cue. Finally, the probe stimulus was always presented at fixation. Thus, lateralization was not confounded by anticipation of a lateralized probe. In sum, the design made sure that retro-cue induced lateralization was indeed reflecting the prioritization of an internal memory representation. That said, a viable interpretation of our findings, and potentially of previous reports of cue-induced alpha-band lateralization, is that lateralization reflects merely a by-product of this prioritization with no direct causal role for memorization.

In line with this interpretation, recent studies have found that alpha-band lateralization in vWM tasks reflects the deployment of spatial attention rather than memory maintenance per se (Hakim et al., 2019; Fukuda, Kang & Woodman, 2016). Wang, Rajsic & Woodman (2019) used pre-cues to indicate the relevant hemifield, and then presented a sequence of to-be-memorized items. While the contralateral delay activity (CDA) of the event-related potential increased with each new item, alpha-band lateralization remained at a constant level, reflecting simply the cued hemifield. Günseli et al. (2019) compared valid and uninformative retro-cues. Interestingly, alpha-band lateralization emerged after retro-cue onset regardless of its validity, unlike the CDA which was reduced for uninformative cues throughout the maintenance interval. Moreover, while the magnitude of the CDA was correlated with accuracy, the magnitude of alpha-band lateralization was not. This indicates that alpha-band lateralization induced by a retro-cue reflects an automatic deployment of spatial attention to a memory representation regardless of its task-relevance. Together with our finding that the extent of alpha-band lateralization is neither correlated with the accuracy of the target representation nor with the suppression of the distractor’s influence, the findings suggest that the deployment of attention reflected by retro-cue induced alpha-band lateralization only coincides with, but is not causally involved in the prioritization yielding the retro-cueing benefit.

Conclusion

Alpha-band lateralization has been suggested as one candidate neuronal mechanism to support the retro-cueing benefit in visual working memory. Our main objective was to test whether this support improves the quality of the memorized target or inhibits the distracting influence of the non-target. We found that it does neither: the data showed a clear retro-cue benefit and a biasing influence of non-target items as well as clear cue-induced alpha-band lateralization, but the magnitude of this lateralization was not correlated with any performance parameter. This finding may indicate that alpha-band lateralization, which is typically observed in response to mnemonic cues, indicates an automatic shift of attention that only coincides with, but is not directly involved in mnemonic prioritization.

Supplemental Information

Supplemental Information 1 Supplemental analyses.

(S1) Comparison of classification accuracy with a conventional set of electrodes. (S2) Alternative analysis of the correlation between individual behavioral retro-cueing benefits and average alpha oscillation. (S3) Interaction of memory bias and target-distractor similarity. (S4) Reliability of lateralization and behavioral measures.

Click here for additional data file.

Supplemental Information 2 Comparison of custom equidistant and international 10/20 EEG montages.

Top: International 10/20 EEG montage. Bottom: Custom "Easycap M34" montage with equidistant positioning of electrodes. The scales of both plots are matched.

Click here for additional data file.

We thank Dr. Daniel Kaiser for helpful discussions. We also thank Davina Hahn and Johanna Rehder for help during data acquisition.

Additional Information and Declarations

Competing Interests

Author Contributions

Human Ethics

Data Availability

1 Note that we conducted a similar analysis using a simpler and more “conventional” approach, in which we correlated participants’ behavioral retro-cue benefits to their alpha-band lateralization, quantified as the difference between ipsilateral and contralateral electrodes. This analysis is reported in Supplemental Analysis S2.

The authors declare that they have no competing interests.

Wanja A. Mössing conceived and designed the experiments, performed the experiments, analyzed the data, prepared figures and/or tables, authored or reviewed drafts of the paper, and approved the final draft.

Niko A. Busch conceived and designed the experiments, authored or reviewed drafts of the paper, and approved the final draft.

The following information was supplied relating to ethical approvals (i.e., approving body and any reference numbers):

The ethics committee of the Faculty of Psychology and Sports Sciences, University of Münster, granted Ethical approval to carry out the study within its facilities (Ethical Approval Ref: 2016-23-NB).

The following information was supplied regarding data availability:

Data and code for all analyses are available at OSF: Mössing, Wanja A, and Niko Busch. 2020. “Retro-Cues and Alpha (Mössing & Busch, 2020).” OSF. April 24. osf.io/r5ksd.

Please refer to the included README.md for instructions on how to use this repository.

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
