# Peer review of "Lateralized alpha oscillations are irrelevant for the behavioral retro-cueing benefit in visual working memory"

_PeerJ, doi:10.7717/peerj.9398_

## Round 0.1 · original submission · Major Revisions

Two experts in the field have reviewed your manuscript. One is quite positive about the potential for publication, and the other cites a set of serious concerns that need to be addressed before the manuscript is suitable for publication. I share the concerns of Reviewer 2, who has provided parsimonious alternative explanations for the null effects you observed...Explanations that disallow adjudication between your competing hypotheses.

You have a daunting task ahead of you. Unless the critiques from Reviewer 2 are thoroughly addressed, I cannot promise that this manuscript will be acceptable for publication upon revision. However, since both reviewers spoke positively about the design and reporting of your research, I wanted you to have an opportunity to attempt a revision.

Beyond the critiques from the reviewers, I have one additional request: Please add a statement to the paper confirming whether you have reported all measures, conditions, data exclusions, and how you determined sample sizes. You should, of course, add any additional text to ensure the statement is accurate. This is the standard reviewer disclosure request endorsed by the Center for Open Science [see http://osf.io/project/hadz3]. I include it in every review.

Reviewer 1 ·

Basic reporting

This is a very nice manuscript that will make an excellent addition to the literature. It is written very clearly and concisely.

Experimental design

The study has solid methods and analysis approach.

Validity of the findings

I only have minor comments, which do not dampen my enthusiasm.

Additional comments

The study tested the hypothesis that the magnitude of lateralized alpha oscillations affects the retro-cue benefit in working memory. The authors found that alpha lateralization magnitude was not correlated with this benefit. This is a very nice manuscript that will make an excellent addition to the literature. It is written very clearly and concisely. The study has solid methods and analysis approach.

I only have minor comments, which do not dampen my enthusiasm.

With regard to the behavioral data, the authors show that retro-cues increased memory precision as compared to neutral cues and that there was an attractive bias of the target item towards the non-target item regardless of the cueing condition. These results are interesting but also somehow surprising. First, most studies on retro-cue benefit that used mixture modeling to estimate the precision and guessing parameter found consistently benefits in guessing but only mixed evidence for benefits in precision (e.g. Souza&Oberauer, 2016). This mixed evidence is in contrast to the "extreme" evidence in the present study. Second, when two items are encoded simultaneously, most studies have reported a repulsive bias (e.g. Blakemore et al., 1970; Sweeny et al., 2011; Czoschke et al., 2019) rather than an attractive bias. It would be great if the authors could comment on these discrepancies.

A strength of the study is its sophisticated analysis approach that included a correlation of single-trial alpha lateralization (via decoding) with single-trial behavioral performance. However I wonder whether a much simpler approach, e.g. correlation of individual retro-cue benefits with individual magnitudes of lateralized alpha (or even by splitting subjects into two groups based on the strength of behavioral benefit and comparing the alpha power), would also be useful and would produce similar results?

The final point is somewhat pedantic: Using the cluster-based permutation test (CBPT) the authors report that „alpha-band power was stronger at electrodes located ipsilateral compared to contralateral relative to the cued target in the time range from 300 to 1070 ms over posterior electrodes“. While this cluster permutation test allows statements about significant differences between the ipsilateral versus contralateral condition, it does not allow statements about the specific frequency (alpha), timing (300-1070) or electrode position (posterior electrodes) of this effect. This is because no statistical inference was made about individual data points (frequency, time, electrode position). Please see e.g. http://www.fieldtriptoolbox.org/faq/how_not_to_interpret_results_from_a_cluster-based_permutation_test/ or Sassenhagen & Draschkow (2019). That means that to prove that the observed effect was present, e.g., at alpha frequency, requires a different statistical approach. Alternatively and more simply, the original statement could be reframed.

Reviewer 2 ·

Basic reporting

Reporting in the manuscript is excellent.

Experimental design

The experimental design is very good. As it pertains to the correlational analyses (does variation in lateralized alpha predict memory performance on cued trials), the ease of the memory task may be a hinderance to detecting a true relationship.

Validity of the findings

Very well-conducted analyses, but null results may not reflect a true lack of a connection between attention-to-memory and lateralized alpha, as discussed further in the General Comments section. Although a null relationship between strong/weak lateralized alpha trials or participants and memory performance is consistent with lateralized alpha not being related to memory prioritization, it could also stem from low variation in successfully orienting attention to the right memory, power reduction by median splitting, noisy EEG classification used to predict behavioural performance (as opposed to noiseless trial labels used in the cue/no-cue contrast), or that differences in power (not topography) index attention-to-memory.

Additional comments

This study poses a timely and important question: are alpha-oscillations a true neural correlate for the attention-to-memory process studied using retro-cues? The authors replicate previous findings of alpha lateralization when lateralized cues orient attention to locations of relevant items in memory but asked further whether variation in alpha lateralization can be used to predict variations in measures of attention-to-memory on a trial-by-trial, or participant-by-participant basis. Failing to find successful predictions, the authors conclude that it is unlikely that alpha lateralization marks the actual prioritization of memories.

I am generally favourable to the approach taken here, and the authors should be commended for conducting such high-quality research, but I have reservations about whether this approach provides strong evidence for the authors’ conclusions. Although the conclusions are plausible (variability in memory performance on retro-cue trials could be a matter losing information, not failing to try and access a memory, the latter of which could be what lateralized alpha indicates), I am concerned that these null findings could just as well be explained other ways, such as the inherent noisiness of EEG analyses, potentially low reliable variation in the memory benefits from retro-cues, or that amplitude differences (not topographical changes) reflect attention-to-memory. While the present results do suggest that this particular method of quantifying lateralized alpha variation isn’t a very good way of discriminating when or for whom retro-cues improve memory in this particular task, it does not follow that differences in alpha lateralization more broadly are not generated by a neural process that is involved in prioritizing memory representations. Below are some specific concerns that should help to explain my scepticism.

1. A small behavioural retro-cuing effect. The concentration contrast between no-cue and cue, while significant, is modest in that even no-cue trials exhibit very high precision (>25) and very low guessing, meaning that memory is always quite good, but just a little better with a cue. Given this very high degree of memory accuracy, ceiling effects a bit of a concern, and indeed the retro-cuing effect has only small-medium effect sizes for all affected memory parameters (d = 0.35 - 0.4). It’s possible that a larger retro-cuing effect (potentially achieved by increasing the number of to-be-remembered objects) could reveal a connection between lateralized alpha and performance on cued trial.
2. A need to establish meaningful variability in behavioural retro-cuing effects. To convincingly argue that lateralized alpha oscillations are not actually related by directing attention to a memory, it is necessary to establish that there is a healthy amount of variability (trial-to-trial or individual-to-individual) in attention to memory that can be reliably measured behaviourally. This is important, as it could be that attention-to-memory in these data are highly consistent across people and trials, which would permit measurement of an average difference in lateralized alpha on cued trials rather than uncued trials, but would not permit not the detection of covariation in memory performance on cued trials with lateralized alpha on cued trials. To do this, one has to separate measurement error from true variability caused by the latent process (attention-to-memory) to quantify the latter. Unfortunately, there does not seem to be an opportunity to assess the reliability of trial-to-trial variability in the retro-cuing, but perhaps some estimate of individual-to-individual reliability could be estimated using data from the separate sessions?
3. Median splitting dilutes power. Although median splits are simple to implement, they are not the best approach when sensitivity needs to be maximized. Median splits mean that observations (whether they are trials or observers) that fall in the mid-range of a distribution are included in both low- and high- groups, diluting the influence of the extreme scores that would drive a difference.
4. Imperfect classification. While being able to correctly classify the direction of a retro-cue from EEG on average 61% of the time is impressive, it does suggest there is a considerable amount of noise in the signal being used to predict behavioural performance. Surely, it is not expected that on trials that are mis-classified, participants are actually attending to the wrong hemifield, leading to classification in the opposite direction of that trial’s label. Instead, if ~40 of trials are incorrect guesses, then shouldn’t ~40% be correct guesses as well (given a left/right choice)? This would mean that on average, only about 20% of classifications can be confidently attributed to signal. This is good enough for establishing a non-zero relationship between lateralized alpha and the retro-cue (note that there is zero noise in the condition labels being predicted, as they are experimenter-determined), but probably not enough for establishing the absence of a relationship between lateralized alpha and performance (which certainly contains variability), unless the to-be-predicted effect is large to make up for the noise in the predictor variable. It is also worth noting that the direction of the participant-to-participant differences are in the right direction (shallower slopes for concentration effect for bad than good participants, less mean bias as well).
5. Specific approach. To be convinced that lateralized alpha has truly no connection to attention-to-memory, it is also important to demonstrate that other methods of evaluating this relationship also fail. The authors have tested one of what seem quite a few methods of quantifying lateralized alpha, finding no evidence it can predict memory performance on retro-cue trials. Perhaps that conclusion is specific to this approach? In particular, this quantification of lateralized alpha relies specifically on variation in its topography. A simpler approach would be to compare whether variability in the overall ipsi-contra alpha power instead predicts the behavioural measures of attention-to-memory. This is especially worth considering given that it is the variability in overall lateralized alpha power (not topography) that is associated with the retro-cuing effect (here and elsewhere).

One minor query: the authors report a 2x2 (good/weak trial X cue/no-cue) ANOVA in their between-trials alpha analysis, but I am not sure how good/bad lateralization was determined for no-cue trials. Can this be clarified?

---

## Round 0.2 · accepted · Accept

As you can see, both reviewers were satisfied that their critiques have been addressed by your revision. I am impressed by your thorough and thoughtful responses to their feedback. Bravo!

Reviewer 1 ·

Basic reporting

no comments

Experimental design

no comments

Validity of the findings

no comments

Additional comments

The authors did a great job replying to my comments. I have no further comments.

Reviewer 2 ·

Basic reporting

Already good.

Experimental design

Already good.

Validity of the findings

Improved communication.

Additional comments

The authors have done an excellent job addressing my concerns; I admit that I am pleasantly surprised how much they were able to adequately address. I appreciate their sensitivity in interpreting null results, and the further work they have done to consider and address alternative explanations. Having demonstrated good reliability and variability in both measures makes the null findings quite a bit more convincing. As such, I now feel this paper will be an excellent addition to the literature and stimulate useful experimentation and theorizing about the cognitive function of lateralized posterior alpha.